

# Arithmetic processing in children with dyscalculia: an event-related potential study

Sonia Y. Cárdenas[1], Juan Silva-Pereyra[2], Belén Prieto-Corona[2], Susana A. Castro-Chavira[1] and Thalía Fernández[1]

[1] Departamento de Neurobiología Conductual y Cognitiva, Instituto de Neurobiología, Universidad Nacional Autónoma de México, Querétaro, México
[2] Facultad de Estudios Superiores Iztacala, Universidad Nacional Autónoma de México, Tlalnepantla, Estado de México, México

## ABSTRACT

**Introduction:** Dyscalculia is a specific learning disorder affecting the ability to learn certain math processes, such as arithmetic data recovery. The group of children with dyscalculia is very heterogeneous, in part due to variability in their working memory (WM) deficits. To assess the brain response to arithmetic data recovery, we applied an arithmetic verification task during an event-related potential (ERP) recording. Two effects have been reported: the N400 effect (higher negative amplitude for incongruent than for congruent condition), associated with arithmetic incongruency and caused by the arithmetic priming effect, and the LPC effect (higher positive amplitude for the incongruent compared to the congruent condition), associated with a reevaluation process and modulated by the plausibility of the presented condition. This study aimed to (a) compare arithmetic processing between children with dyscalculia and children with good academic performance (GAP) using ERPs during an addition verification task and (b) explore, among children with dyscalculia, the relationship between WM and ERP effects.

**Materials and Methods:** EEGs of 22 children with dyscalculia (DYS group) and 22 children with GAP (GAP group) were recorded during the performance of an addition verification task. ERPs synchronized with the probe stimulus were computed separately for the congruent and incongruent probes, and included only epochs with correct answers. Mixed 2-way ANOVAs for response times and correct answers were conducted. Comparisons between groups and correlation analyses using ERP amplitude data were carried out through multivariate nonparametric permutation tests.

**Results:** The GAP group obtained more correct answers than the DYS group. An arithmetic N400 effect was observed in the GAP group but not in the DYS group. Both groups displayed an LPC effect. The larger the LPC amplitude was, the higher the WM index. Two subgroups were found within the DYS group: one with an average WM index and the other with a lower than average WM index. These subgroups displayed different ERPs patterns.

**Discussion:** The results indicated that the group of children with dyscalculia was very heterogeneous and therefore failed to show a robust LPC effect. Some of these children had WM deficits. When WM deficits were considered together with dyscalculia, an atypical ERP pattern that reflected their processing difficulties

Corresponding author
Thalía Fernández,
thaliafh@yahoo.com.mx

emerged. Their lack of the arithmetic N400 effect suggested that the processing in this step was not useful enough to produce an answer; thus, it was necessary to reevaluate the arithmetic-calculation process (LPC) in order to deliver a correct answer.

**Conclusion:** Given that dyscalculia is a very heterogeneous deficit, studies examining dyscalculia should consider exploring deficits in WM because the whole group of children with dyscalculia seems to contain at least two subpopulations that differ in their calculation process.

## INTRODUCTION

According to the Diagnostic and Statistical Manual of Mental Disorders, 5th Edition (DSM-5; *American Psychiatric Association, 2013*), dyscalculia refers to difficulties with number sense, number facts, and calculation (i.e. having a poor understanding of numbers, their magnitudes and relationships, counting on fingers to add single-digit numbers instead of recalling math facts as peers do, becoming lost in the midst of arithmetic computation and switching procedures). The academic skills of children with dyscalculia are substantially below those expected for their chronological age, which can cause significant difficulties in academic performance and in activities of daily living (*American Psychiatric Association, 2013*). Dyscalculia cannot be better accounted for by intellectual disabilities, uncorrected visual or auditory acuity, other mental or neurological disorders, psychosocial adversity, lack of proficiency in the language of academic instruction, or inadequate educational instruction (*American Psychiatric Association, 2013*).

Dyscalculia is a heterogeneous cognitive disorder (*Kaufmann et al., 2013*). A known source of this heterogeneity is working memory (WM), which varies markedly between children with dyscalculia (*Andersson & Lyxell, 2007*; *Geary, 1993*; *Mammarella et al., 2017*). The WM system provides online storage of information and its subsequent manipulation through four subsystems: the phonological loop, the visuospatial sketchpad, the episodic buffer, and the central executive (*Baddeley, 2006*). In the domain of mathematics, the phonological loop holds intermediate arithmetic results in the form of linguistic information, and plays a role in mathematical abilities that involve the articulation of numbers, such as counting, problem-solving, and arithmetic fact retrieval (*Geary, 1993*; *Shen, Liu & Chen, 2018*). The visuospatial sketchpad supports the construction of visual representations of numerical information and is, thus, related to spatial aspects of calculation, such as decomposition strategies (*Foley, Vasilyeva & Laski, 2017*; *Simms et al., 2016*). The episodic buffer provides a temporary storage that links information from the two slave subsystems and long-term memory, allowing the maintenance of multi-code number representations (*Camos, 2018*). Finally, the central executive coordinates and monitors simultaneous processing and keeps track of math tasks that have already been performed (*DeStefano & LeFevre, 2004*; *Fuchs et al., 2005*; *Holmes &*

*Adams, 2006*). Children with dyscalculia may show difficulty in verbal short-term memory and verbal WM (*Attout & Majerus, 2015*; *Berninger, 2008*; *Hitch & McAuley, 1991*; *Peng & Fuchs, 2016*; *Shen, Liu & Chen, 2018*; *Swanson & Siegel, 2001*), visuospatial short-term memory and visuospatial WM (*McDonald & Berg, 2018*; *Mammarella et al., 2017*; *Rotzer et al., 2009*; *Schuchardt, Maehler & Hasselhorn, 2008*), and the central executive (*Andersson & Lyxell, 2007*; *Meyer et al., 2010*; *Vanbinst & De Smedt, 2016*). In addition, these children have been reported to show a slower processing speed (*Geary, Hoard & Hamson, 1999*; *Landerl, Bevan & Butterworth, 2004*; *Shalev, Manor & Gross-Tsur, 2005*).

Behavioural performance (accuracy and response time) in arithmetic tasks depends on the arithmetic ability of the subject (*Cipora & Nuerk, 2013*; *LeFevre & Kulak, 1994*; *Núñez-Peña & Suárez-Pellicioni, 2012*) as well as individual characteristics such as age (*De Smedt et al., 2009*; *Geary & Wiley, 1991*; *Geary, Bow-Thomas & Yao, 1992*) and school grade (*Geary, 2004*; *Imbo & Vandierendock, 2008*). Behavioural performance also depends on the task features. In an arithmetic verification task, in which the arithmetic operation (context) is followed by a possible solution (probe) that may or may not match the correct result of the operation, the priming phenomenon manifests as a shorter response time in the presence of facilitation provided by the context, that is when the probe digit coincides with the result of the proposed arithmetic operation (congruent condition). One explanation for this phenomenon is that the congruent solution is more quickly recovered from memory (*Niedeggen & Rösler, 1999*; *Niedeggen, Rösler & Jost, 1999*). Thus, to provide a correct answer, a child needs to perform adequate arithmetic processing (to choose the correct probe) as well as adequately maintain the result in verbal WM via the verbal short-term memory, which leads to facilitation.

All previously mentioned studies used behavioural variables to draw their conclusions. Although behavioural assessments of performance during tasks can yield data for variables such as response time and percentage of correct answers, they ignore the subject's cerebral and cognitive processes. In contrast, the high temporal resolution of event-related potentials (ERPs) can elucidate these processes by representing the processing through each millisecond and thereby allowing chronologic analysis of brain function through different cognitive processes.

Event-related potentials have previously been used to study arithmetic processing. Evaluation of arithmetic verification processing in healthy young adults by using ERPs reported a negative wave with greater amplitude in an incongruent condition (i.e. when there is no facilitation provided by the context) than in a congruent condition (i.e. with facilitation provided by the context) (*Dong et al., 2007*; *El Yagoubi, Lemaire & Besson, 2003*; *Hinault & Lemaire, 2016*; *Prieto-Corona et al., 2010*; *Szűcs & Csépe, 2005*). The arithmetic N400 component is a negative waveform that begins at around 250 ms, peaks at around 400 ms, and is maximal over the centroparietal area on the scalp (*Dickson & Federmeier, 2017*; *Hinault & Lemaire, 2016*; *Jost, Hennighausen & Rösler, 2004*; *Niedeggen, Rösler & Jost, 1999*; *Niedeggen & Rösler, 1999*; *Prieto-Corona et al., 2010*). If the arithmetic N400 component in the incongruent is larger than in the congruent condition, the significant difference in amplitude between these components is known as the arithmetic N400 effect, which reflects the strength of the probe's relationship with the

context (i.e. arithmetic operation) (*Niedeggen & Rösler, 1999*). N400 is thought to reflect the automatic retrieval of arithmetic facts from long-term memory and its magnitude represents the strength of the probe's relationship with the context (i.e. arithmetic operation) (*Niedeggen & Rösler, 1999*). However, lack of concordance between automatic recovery of the correct results and the probe may involve additional inhibitory processes (*Hinault & Lemaire, 2016*).

Studies with different populations have indicated that the arithmetic N400 effect is modulated by arithmetic abilities. For example the effect is greater in adults or teenagers with better arithmetic abilities than in adults or teenagers, respectively, with poorer arithmetic abilities (*Núñez-Peña, Gracia-Bafalluy & Tubau, 2011*; *Núñez-Peña & Suárez-Pellicioni, 2012*, *2015*; *Soltész et al., 2007*; *Soltész & Szűcs, 2009*; *Thevenot, Fanget & Fayol, 2007*). Comparisons between children and adults have revealed differences in latency and topographical distributions of the arithmetic N400 effect (*Prieto-Corona et al., 2010*). Further, younger children show longer latencies than older children (*Dong et al., 2007*).

Another ERP component that has been used to investigate arithmetic processing in adults and children is the late positive component (LPC). This follows the arithmetic N400 component, appearing between 500 and 700 ms. The LPC is a positive deflection in the ERP waveform that shows a parietal (*Jasinski & Coch, 2012*; *Niedeggen, Rösler & Jost, 1999*; *Núñez-Peña & Suárez-Pellicioni, 2015*; *Xuan et al., 2007*) or centro-parietal (*Núñez-Peña & Escera, 2007*; *Núñez-Peña & Suárez-Pellicioni, 2012*; *Prieto-Corona et al., 2010*) topography, mainly over the right hemisphere (*Jasinski & Coch, 2012*; *Niedeggen & Rösler, 1999*; *Niedeggen, Rösler & Jost, 1999*). The significant difference in amplitudes between LPC components elicited by incongruent and congruent conditions is known as the LPC effect when amplitude in an incongruent condition is larger than in a congruent condition (*Jost, Hennighausen & Rösler, 2004*; *Niedeggen, Rösler & Jost, 1999*; *Núñez-Peña & Suárez-Pellicioni, 2012*; *Prieto-Corona et al., 2010*; *Szűcs & Csépe, 2005*; *Szűcs & Soltész, 2010*). The LPC effect is associated with processing re-evaluation (*Núñez-Peña & Suárez-Pellicioni, 2012*; *Prieto-Corona et al., 2010*; *Szűcs & Soltész, 2010*), and its amplitude is modulated by the plausibility of a presented condition (*Niedeggen & Rösler, 1999*; *Núñez-Peña & Escera, 2007*; *Núñez-Peña & Honrubia-Serrano, 2004*; *Núñez-Peña & Suárez-Pellicioni, 2015*; *Szűcs & Soltész, 2010*). Some authors have proposed that the LPC effect reflects surprise due to an out-of-context stimulus (*Donchin & Coles, 1997*; *Núñez-Peña & Suárez-Pellicioni, 2012*; *Polich, 2007*). The LPC effect is greater in adults than in children (*Zhou et al., 2011*) and in individuals with better arithmetic abilities than in those with arithmetic deficits (*Iguchi & Hashimoto, 2000*; *Núñez-Peña, Gracia-Bafalluy & Tubau, 2011*; *Núñez-Peña & Honrubia-Serrano, 2004*; *Núñez-Peña & Suárez-Pellicioni, 2012*, *2015*; *Szűcs & Soltész, 2010*).

In summary, children with dyscalculia may show deficits in WM in addition to the characteristic mathematical problems, making them a heterogeneous group. Although ERPs have shown that neural processing in these children differs from that in children with typical abilities, the effects of an additional WM deficit on the processing of an arithmetic verification task at the neural level remain unknown. Since ERPs can reveal or highlight mechanisms that remain undetected by behavioural measures, the body of knowledge

about dyscalculia may be enhanced by comparing ERPs of children with dyscalculia and those with typical development while the children perform an arithmetic verification task. Thus, the first aim of the current study was to compare the arithmetic processing between children with dyscalculia and children with good academic performance (GAP) by assessing their ERPs during an addition verification task. The second aim was to explore the relationship between WM and ERPs in children with dyscalculia. We hypothesised that, in comparison with children with GAP, children with dyscalculia would show (1) less accurate or slower behavioural responses on an arithmetic verification task, (2) smaller or later arithmetic N400 and LPC effects, and (3) poorer performance on WM tests. In addition, we explored the possibility of a relationship between WM performance and the N400 and LPC effects in children with dyscalculia.

## METHODS

### Ethics

This research was conducted in accordance with the ethical principles of the Declaration of Helsinki. The Bioethics Committee of the Neurobiology Institute at the Universidad Nacional Autónoma de México approved the experimental protocol (INEU/SA/CB/145). Children and their parents gave written informed consent to participate in this study.

### Participants

Forty-four right-handed children aged between 9 and 11 years participated in this study. The participants were selected from a sample of 167 children from public and private elementary schools in Querétaro, México. The study was carried out in 2015–2016. The interview, examinations and psychological and neuropsychological tests were administered around 2 months before the ERPs. After completing a semi-structured interview, we excluded 16 children due to low socioeconomic status (the mother had not completed elementary school and/or per capita income was less than 100% of the minimum wage; *Harmony et al., 1990*) and two children who presented with epilepsy. In addition, we excluded six children with intellectual disability (i.e. IQ < 70; Wechsler Intelligence Scale for Children, 4th Edition, Spanish version; *Wechsler, 2007*), 52 children who showed psychiatric disorders (i.e. ADHD, behaviour disorder, and/or oppositional defiant disorder as identified with MiniKid (*Ferrando et al., 1998*) and neuropsychiatric assessments), and two children with uncorrected hypoacusis were excluded as well.

The remaining 89 children completed the arithmetic subtest of the Child Neuropsychological Assessment (*Matute et al., 2005*), which is standardised and includes norms for the Mexican population. Its arithmetic domain consists of three subdomains (counting, number management and calculus). Thirty participants who performed at or below the 9th percentile in at least one arithmetic subdomain were assigned to a group of children with dyscalculia (DYS group), and 28 participants at or above the 37th percentiles in all subdomains were assigned to a group with GAP group. The remaining 31 participants that did not belong to either of these two groups were excluded. Of the selected children, five from the DYS group and two from the GAP group were excluded because their correct answers were below the chance level (58%). Another three children

from the DYS group and four from the GAP group were later excluded due to poor ERP data (see the ERP section below). Thus, the DYS and GAP groups were each represented by 22 participants (11 and 14 girls in the DYS and GAP groups, respectively). The groups did not differ in age, gender ($\chi^2(1) = 0.834$, $p = 0.361$), or monthly family income per capita.

Both groups underwent assessments for the four neuropsychological indices of the Wechsler Intelligence Scale for Children: verbal comprehension index, WM index, processing speed index, and perceptual reasoning index. The children in the GAP group had scores of 85 or higher in all indices, while those in the DYS group showed significantly lower scores on all the indices except the processing speed index, as shown in Table 1. Figure 1 shows the boxplots of the arithmetic subtests of the Child Neuropsychological Assessment and the WM index of the Wechsler Intelligence Scale for Children. All participants had normal or corrected-to-normal visual acuity, and they did not present any history of neurological or psychiatric disorders. Children from both groups were selected from the same schools and were therefore from the same educational environments.

## Stimuli

Each trial of the task started with a warning stimulus (a right-pointed arrow), which was followed by an addition operation with two single-digit operands between 1 and 9. Each addition operation combined the two Arabic digits using the plus sign (+), resulting in 81 different addition operations. Every operation was presented once with each of the correct and incorrect results (congruent and incongruent conditions). The incorrect result was constructed by either adding 2 to the correct result (for 41 facts) or by subtracting 2 from it (for the remaining 40 facts).

## Arithmetic verification task

Figure 2 illustrates the time chart of the task. In each of the 162 trials, a white warning stimulus was presented at the centre of the black screen for 200 ms, followed by a black screen that lasted for 300 ms. A white addition operation then appeared for 1,500 ms, followed by another black screen for 1,500 ms. Subsequently, a white number (probe stimulus) was presented for 1,000 ms on a black screen, which either did or did not match the sum of the numbers (for the congruent or incongruent conditions, respectively). Finally, a black screen was presented for 500 ms. Half of the trials were congruent and half incongruent. Trials were randomised and delivered by Mind-Tracer 2.0 software (Neuronic Mexicana, S.A.; Mexico City, Mexico).

## Procedure

Children were seated in a comfortable chair 70 cm from the computer screen in a sound-attenuated dimly-lit Faraday recording chamber. The experiment began after a training period to familiarise the children with the task, which consisted of 16 trials with feedback. This was followed by 162 trials divided into four blocks (two with 40 and two with 41 trials). Blocks were separated by 1-min rest periods.

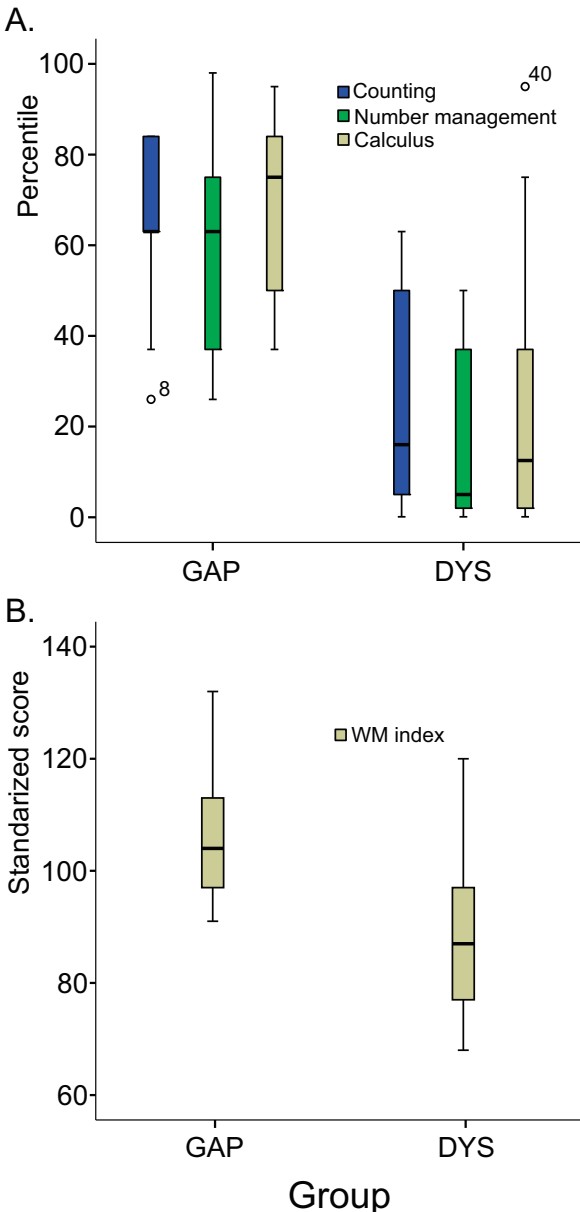

**Figure 1 Variability of arithmetic subdomains and WM index in both groups.** (A) Box-and-whisker plots of the subdomains (counting, number management, and calculus) of the arithmetic subtest of the Child Neuropsychological Assessment in both groups of children (GAP and DYS). (B) Box-and-whisker plots of the working memory index of the Wechsler Intelligence Scale for Children, 4th Edition, Spanish version. The error bars represent the standard deviation.

All children were instructed to relax and maintain their gaze towards the centre of the screen and to avoid blinking when the probe stimulus appeared. They were asked to blink after the response was given, just before the warning stimulus. The children were instructed to respond as quickly and accurately as possible when the probe stimuli were presented. Half the children were instructed to press the mouse key with the right thumb if they thought the probe was correct (congruent condition) and with the left thumb if they

**Table 1 Differences between groups in terms of demographic features, arithmetic performance and intelligence quotient.**

| | Mean ± SD | | t(42) | p-Value |
|---|---|---|---|---|
| | GAP | DYS | | |
| Age (years) | 9.41 ± 1.182 | 9.77 ± 0.813 | −1.189 | 0.241 |
| Monthly economic income | 2421 ± 1188.81 | 1983 ± 1209.76 | 1.211 | 0.233 |
| ENI subscales (arithmetic domain) | | | | |
| Counting | 65.27 ± 15.03 | 24.55 ± 23.86 | 6.772 | <0.001 |
| Number management | 59.73 ± 23.03 | 13.91 ± 17.68 | 7.400 | <0.001 |
| Calculus | 68.18 ± 20.99 | 20.91 ± 26.08 | 6.622 | <0.001 |
| Logical mathematical reasoning | 68.45 ± 22.28 | 51.41 ± 35.02 | 1.926 | 0.061 |
| WISC-IV indexes | | | | |
| Intelligence quotient | 105.95 ± 10.86 | 89.55 ± 10.82 | 5.017 | <0.001 |
| Verbal comprehension index | 102.41 ± 20.69 | 86.23 ± 21.62 | 2.536 | 0.015 |
| Perceptual reasoning index | 100.59 ± 18.71 | 85.68 ± 20.83 | 2.947 | 0.017 |
| Working memory index | 104.23 ± 11.48 | 89.36 ± 13.85 | 3.875 | <0.001 |
| Processing speed index | 97.95 ± 20.71 | 91.68 ± 14.32 | 1.168 | 0.249 |
| WISC-IV subtests | | | | |
| Similarities | 10.77 ± 2.72 | 8.59 ± 4.22 | 2.035 | 0.048 |
| Vocabulary | 11.64 ± 2.82 | 8.55 ± 2.24 | 4.024 | <0.001 |
| Comprehension | 10.82 ± 3.11 | 8.09 ± 2.87 | 3.019 | 0.004 |
| Block design | 10.86 ± 2.76 | 8.68 ± 2.62 | 2.684 | 0.010 |
| Picture Concepts | 10.59 ± 2.70 | 8.86 ± 2.66 | 2.137 | 0.038 |
| Matrix reasoning | 10.45 ± 2.28 | 8.41 ± 1.91 | 3.217 | 0.002 |
| Digit span | 10.73 ± 2.12 | 7.68 ± 2.41 | 4.442 | <0.001 |
| Letter-number sequencing | 11.23 ± 2.20 | 8.14 ± 2.73 | 4.133 | <0.001 |
| Arithmetic | 14.00 ± 3.14 | 9.75 ± 2.86 | 4.417 | <0.001 |
| Coding | 10.23 ± 2.56 | 8.64 ± 1.94 | 2.322 | 0.025 |
| Symbol search | 10.41 ± 2.21 | 9.27 ± 2.18 | 1.711 | 0.094 |

**Note:**
ENI: Child Neuropsychological Assessment (*Matute et al., 2005*); **WISC-IV**: The Wechsler Intelligence Scale for Children, 4th Edition Spanish version (WISC-IV; *Wechsler, 2007*); **GAP**: children with good academic performance; **DYS**: children with dyscalculia.

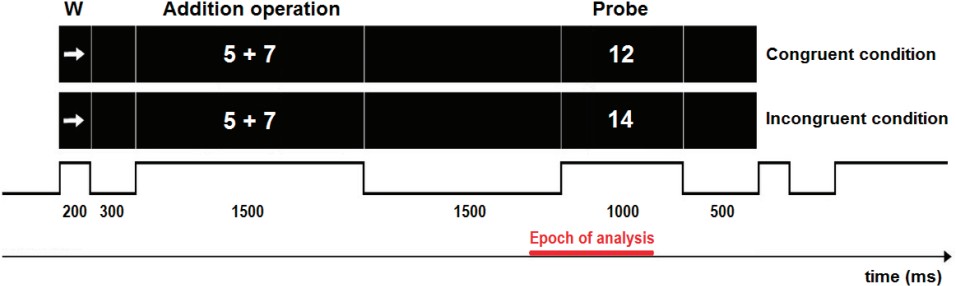

**Figure 2 Depiction of a trial of the addition verification task.** Flowchart of stimuli presentation during individual trials. W, warning stimulus.

thought it was incorrect (incongruent condition). The other half of the children were instructed to do the opposite.

## ERP acquisition and analysis

A 19-channel EEG (Ag/AgCl electrodes held in position with a cap according to the 10–20 International System; Electro-Cap™ International, Inc.; OH, USA), referenced to linked earlobes (A1A2), was recorded using a MEDICID™ IV system (Neuronic S.A.; Mexico City, Mexico) and a Track Walker v5.0 data system while the child was performing the task. The bandwidth of the amplifiers was 0.5–50 Hz, and the sampling frequency was 200 Hz. Impedances in all the recordings were maintained below 5 kΩ. Electro-oculograms were recorded with electrodes located on the superciliary arch and the external canthus of the right eye.

Event-related potentials were computed offline using 1,000 ms EEG epochs from each subject in each experimental condition. The epochs consisted of a baseline period that started 200 ms before the probe onset and ended 800 ms after the probe onset. Baseline correction was performed using the 200 ms pre-stimulus period. An EEG epoch was rejected if visual inspection revealed blinking or ocular movements, electrical activity exceeding 100 microvolts, or amplifier blocking for more than 50 ms at any electrode site. Seven participants (three in the DYS group) had fewer than 20 artifact-free trials per condition, so these participants were excluded. The number of EEG epochs per condition was approximately equal per subject. On average, the DYS and GAP groups had 33 and 39 artifact-free epochs, respectively, for each condition. Accepted EEG epochs associated with correct answers were averaged together to produce one ERP each for the congruent and incongruent conditions for each child. The former was subtracted from the latter (i.e. incongruent minus congruent) to produce one ERP difference wave per child.

## Statistical analysis

### Behavioural data analysis

Statistical analyses of behavioural data were performed using the statistical program SPSS (IBM Statistic 20, Chicago, IL, USA). We conducted mixed 2-way ANOVAs for response times and for correct answers. The percentage of correct answers was transformed by arcsine (square root (percentage/100)) (*Zar, 2010*). Group (GAP, DYS) was included as the between-subjects factor, and condition (congruent, incongruent) was included as the within-subjects factor. The least significant differences method was used for post-hoc pairwise comparisons.

### ERP data analysis

Figure 3 shows the scheme of statistical analyses for the ERP data. All assessments were performed using nonparametric tests with permutations (*Galán et al., 1998*) due to the multiplicity of comparisons and dependent variables and the consequently increased probability of type I errors (*Luck, 2014*). Analyses were carried out using eLORETA software (*Pascual-Marqui et al., 2011*). Five thousand permutations were performed. Global significance for the statistical test (i.e. significant $p$-value level considering all the

**A. Defining analysis time-window**

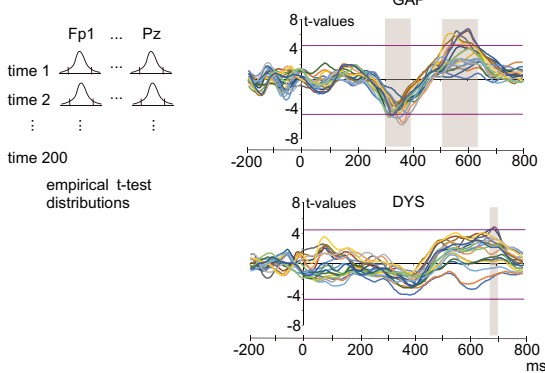

**B. Topographical exploration of ERP effects**

Comparison incongruent vs congruent

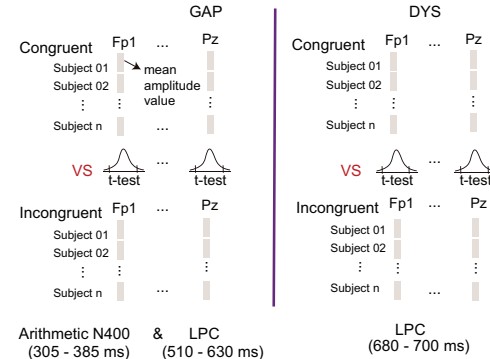

**C. Between-group comparison of ERP effects**

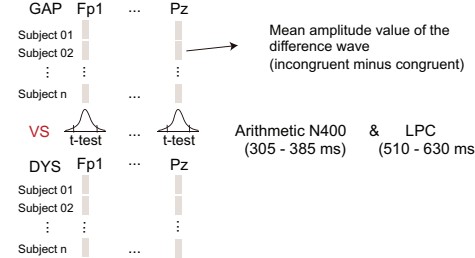

**D. Correlation analyses**

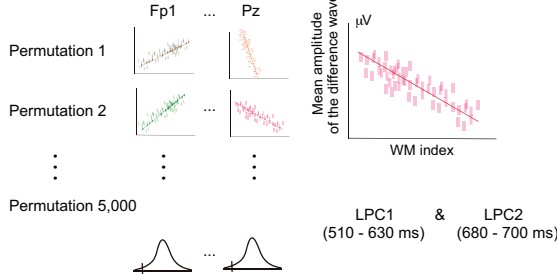

**Figure 3** **Workflow of the statistical analyses of ERP data using non-parametric permutation tests.** (A) Definition of analysed time windows, where significant differences between incongruent and congruent conditions (effects) were evinced; comparison between conditions using multiple *t*-tests, shown at each point of time throughout the electrode sites (colour lines in the coordinate axis). Magenta horizontal

**Figure 3 (continued)**
lines represent the threshold of *t*-values for *p* = 0.05, and grey shadowed boxes represent the analysed time windows where significant differences were found. Colour lines in the coordinate axis represent *t*-values at different electrode sites. (B) Exploration of the topography of ERP effects (incongruent minus congruent) obtained from (A); *t*- tests were computed using the mean amplitude values in each condition for each analysed time window (N400 and LPC in the group GAP and LPC in the group DYS) across all electrode sites. (C) Comparison of the ERP-difference wave between the DYS and GAP groups. Mean amplitude values of the difference waves were used to compute the *t*-tests. (D) Correlation analyses between the working memory index and ERP difference waves for the DYS group, for each electrode site and each ERP window.   

electrodes) was reported as $T_{max}$ and its extreme *p*-value. Because this statistical test is based on an empirical probability distribution, extreme *p*-values were corrected by multiple comparisons.

Time windows of the ERP components are usually defined by the outcomes of previous studies. However, most studies relevant to this experiment tested young adults, who have faster processing than children. To determine appropriate time windows for the arithmetic N400 and LPC effects in children, we performed a non-parametric permutation test to identify significant differences between the ERP waveforms for congruent and incongruent conditions per time point between −200 to 800 ms at all electrode sites (Fig. 3A). In each group of children, we defined the time windows of the arithmetic N400 and LPC.

The next step was to explore the topography of the N400 and LPC effects per group across all electrode sites (Fig. 3B). In addition, eLORETA was used to conduct three analyses that compared the ERP difference waveforms (incongruent minus congruent) between the two groups (GAP, DYS) (Fig. 3C). Five thousand permutations were performed. Significant *t*-values over electrode sites are represented in colour maps (only *t*-values with *p* < 0.05).

We also used eLORETA to perform three correlation analyses in the DYS group between each ERP difference wave form and the WM index across all electrode sites (Fig. 3D). Five thousand permutations were performed. Significance for the statistical test was reported ($r_{max}$ and its extreme *p*-value). Specific significant correlations (*r* value) over electrode sites are represented in colour maps (only *r*-values with *p* < 0.05).

All statistical results for the ERPs were reported taking into consideration all 19 electrodes.

# RESULTS

## Behavioural results

The behavioural results are shown in Fig. 4. The participants in the GAP group showed a significantly higher percentage of correct answers than those in the DYS group ($F_{(1, 42)} = 27.39$, $p < 0.0001$, $\eta_p^2 = 0.395$). The percentage of correct answers in the incongruent condition was significantly higher than that in the congruent condition ($F_{(1, 42)} = 8.67$, $p = 0.005$, $\eta_p^2 = 0.171$), independently of the group. No significant group by condition interaction was noted ($F < 1$).

The responses for all children were significantly faster in the congruent condition than in the incongruent condition ($F_{(1, 42)} = 131.922$, $p < 0.0001$, $\eta_p^2 = 0.759$), but the response

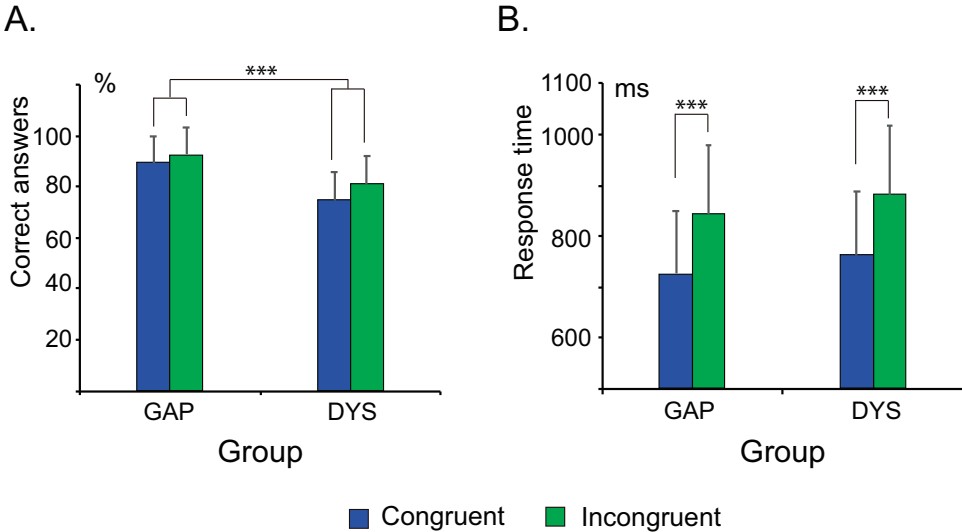

**Figure 4 Behavioural data in both groups of children (GAP and DYS) in the arithmetic verification task.** The correct answer (A) and mean response time (B) in both conditions (congruent and incongruent) and both groups of children. Error bars represent the standard deviation. The DYS group showed a lower percentage of correct answers than the GAP group. ***$p < 0.0001$.

times were not significantly different between the groups ($F < 1$). No significant group by condition interaction ($F_{(1,42)} = 1.114$, $p = 0.297$, $\eta_p^2 = 0.026$) was observed for this assessment. This finding could be attributed to the large age range of the participants, since the automation of solutions to arithmetic problems is a developing process in children of these ages. We tested this possibility by exploring the association between age and response time using Spearman rank correlation analyses within groups. The GAP group showed significant negative correlations for congruent ($r = -0.57$, $p = 0.006$) and incongruent ($r = -0.60$, $p = 0.003$) conditions; however, the DYS group showed no significant correlations for any condition (congruent: $r = -0.29$, $p = 0.195$; incongruent: $r = -0.22$, $p = 0.337$).

## Electrophysiological results
### Time windows for the N400 and LPC effects in the DYS and GAP groups
The statistical results showed significant differences between conditions from 305 to 385 ms and from 510 to 630 ms in the GAP group ($T_{max} = -3.387$, extreme $p = 0.0004$). Figures 5A and 5B show the topography of the significant differences in the first and second windows, which correspond to the arithmetic N400 and LPC effects, respectively, in terms of their latency and polarity (negative and positive, respectively). The LPC effect elicited by the GAP group was named the LPC1 effect. The topographic distribution of both ERP effects corresponds with the findings reported in previous studies in young adults. The arithmetic N400 effect was localised over the frontal midline (*Megías & Macizo, 2016*; *Prieto-Corona et al., 2010*) and left centroparietal area (*Avancini, Galfano & Szűcs, 2014*; *Avancini, Soltész & Szűcs, 2015*; *Dickson & Federmeier, 2017*). The LPC effect was observed over the centro-parieto-temporal area, mainly in the right hemisphere

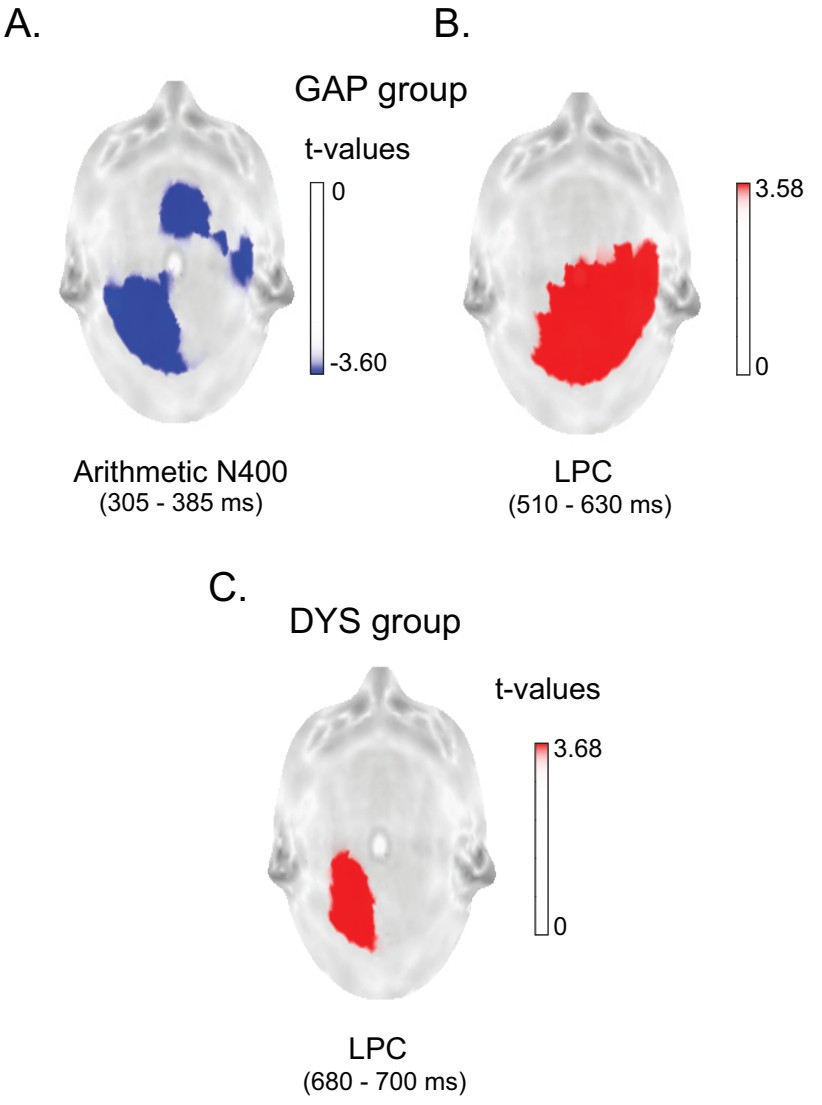

**Figure 5 Statistical parametric maps of the arithmetic N400 and LPC effects in both groups.** Top: GAP group. (A) Differences between conditions at 305–385 ms (arithmetic N400). (B) Differences between conditions at 510–630 ms (LPC effect). Bottom: DYS group. (C) Differences between conditions at 680–700 ms (LPC effect). Blue and red colours represent the *t*-values that were above the threshold of significance (*p* < 0.001). In the GAP group, the arithmetic N400 effect was elicited at P3, O1, T4, T5, Fz, and Pz and the LPC effect was elicited at C4, P4, O1, O2, T4, T6, Cz and Pz, while in the DYS group, the LPC effect was observed at P3 and O1. All *p* < 0.001.

(*Avancini, Soltész & Szűcs, 2015*; *Dickson & Federmeier, 2017*; *Jasinski & Coch, 2012*; *Niedeggen & Rösler, 1999*). In contrast, the DYS group only displayed a significant difference between 680 and 700 ms ($T_{max}$ = 4.84, extreme *p* = 0.021), as shown in Fig. 5C, which could correspond to a late LPC effect (named the LPC2 effect).

The grand averages of the ERPs in the T3 and C3 electrodes in the two task conditions for both groups are shown in Fig. 6. This figure clearly illustrates that the lack of arithmetic N400 effect in the DYS group is not associated with a lack of response, but with similarly large amplitudes for both arithmetic N400 components in each condition.

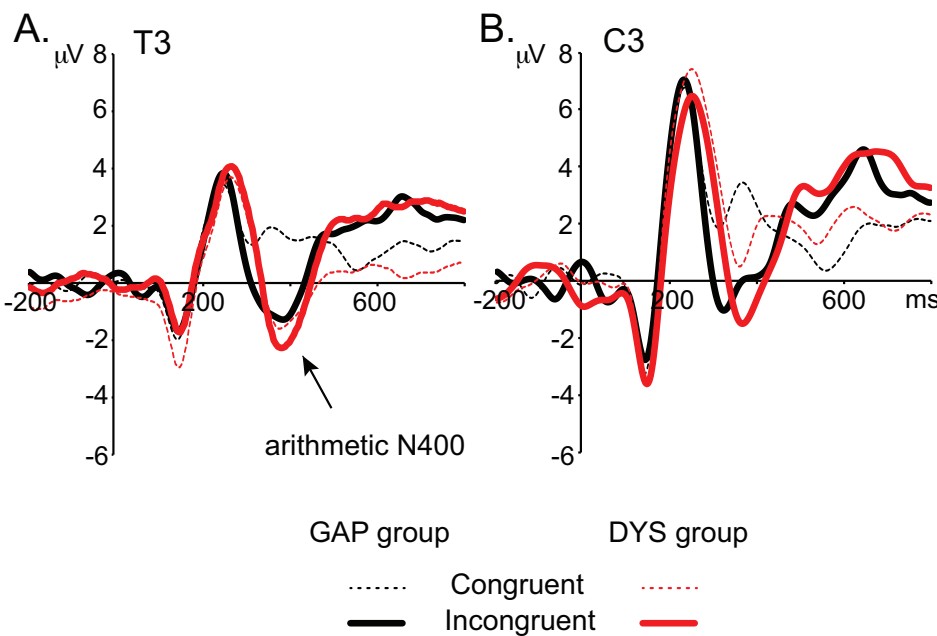

**Figure 6 ERP wave grand averages.** (A) T3 electrode. (B) C3 electrode. The GAP group responses to congruent and incongruent conditions are represented by the black continuous and discontinuous lines, while the DYS group responses to congruent and incongruent conditions are represented by the red continuous and discontinuous lines, respectively. Negativity is plotted downwards.

### ERP difference waveforms in the DYS and GAP groups

Having identified appropriate time windows for the N400 and LPC effects in each group, three statistical analyses for independent samples were performed using the permutation technique (considering all electrodes) to compare the ERP difference waves in the GAP and DYS groups per time window identified (305–385 ms, 510–630 ms and 680–700 ms). The GAP children showed a significantly larger amplitude for the arithmetic N400 effect over T5 ($T_{max} = -3.58$, extreme $p = 0.007$) and a significantly larger LPC1 effect over Fp2 (global $T_{max} = 3.01$, extreme $p = 0.032$) than the DYS children. In the LPC2 time window, no differences between groups were observed ($T_{max} = 1.46$, extreme $p = 0.45$). Figure 7 shows the statistical colour maps of the arithmetic N400 effect and LPC effect comparisons between the two groups (GAP vs. DYS).

### Associations between WM and ERPs

The heterogeneity that characterises behavioural performance in dyscalculia (*Kaufmann et al., 2013*) is also likely reflected on ERPs since ERPs correspond to the brain processing that underlies performance, which indicates that data dispersion is higher in the DYS group. Moreover, the DYS group showed more outliers than the GAP group (Fig. 8). One source of this heterogeneity has been proposed to be WM (*Andersson & Lyxell, 2007*; *Geary, 1993*).

   The children with dyscalculia were assessed according to their WM indices and distributed into two subgroups: one with average WM indices (scores equal to 85 or higher;

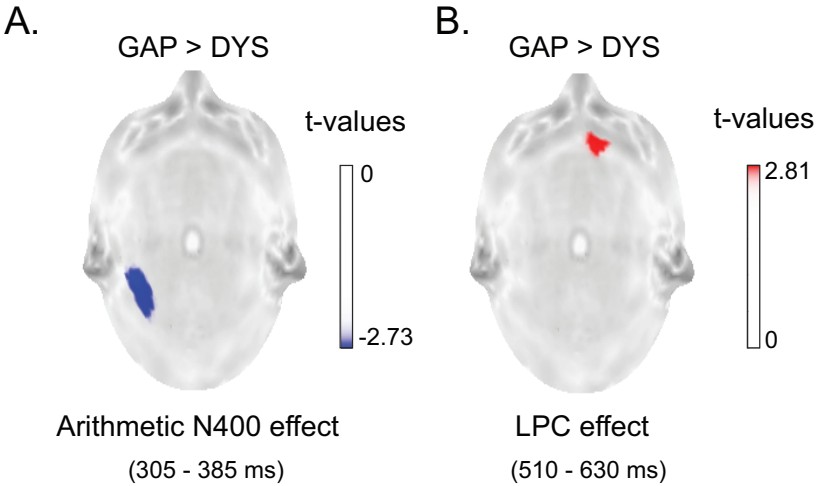

**Figure 7 Differences between groups in arithmetic N400 and LPC effects.** (A) Statistical map of the comparison between groups based on the difference between conditions (incongruent minus congruent) for the arithmetic N400 (305–385 ms) at T5. (B) Statistical map of the comparison between groups based on the difference between conditions (incongruent minus congruent) for the LPC (510–630 ms) at Fp2. The blue and red spots represent significant differences between groups (*t*-values *p* < 0.05).

*n* = 13, six girls) and the other with lower-than-average WM indices (scores < 85; *n* = 8, four girls). Figure 9 displays the grand average of the difference wave for these two subgroups of children with dyscalculia, as well as children with GAP. The children with dyscalculia and a low WM index score seemed to show one N200 peak, one arithmetic N400 peak, and two LPC peaks, representing an atypical ERP pattern for this task. In contrast, children with dyscalculia, but with average WM index scores, showed a similar ERP pattern to children with GAP.

For the children with dyscalculia, correlation analyses between the WM index scores and the amplitude values of the difference wave at each electrode site were performed in every ERP window. No significant correlation was found between the WM index and the difference wave in the N400 window. However, in both LPC windows, significant positive correlations were found between the WM index and LPC difference waves. In the LPC1 time window, a greater WM index correlated with a greater amplitude in the LPC effect over O2 and T6 ($r_{max}$ = 0.68, extreme *p* = 0.0056) and, in the LPC2 time window, a greater WM index correlated with a greater amplitude of the LPC effect over T6 ($r_{max}$ = 0.61, extreme *p* = 0.0178). Figure 10 shows statistical colour maps for the correlations between the WM index and the LPC effects.

## DISCUSSION

The first objective of this study was to compare arithmetic verification processing in children with dyscalculia with that in children with GAP during an addition verification task by using ERPs. To our knowledge, this is the first study to compare the ERPs of these two populations of children. We expected poorer behavioural performance (lower percentage of correct answers and/or longer response times) in the children with

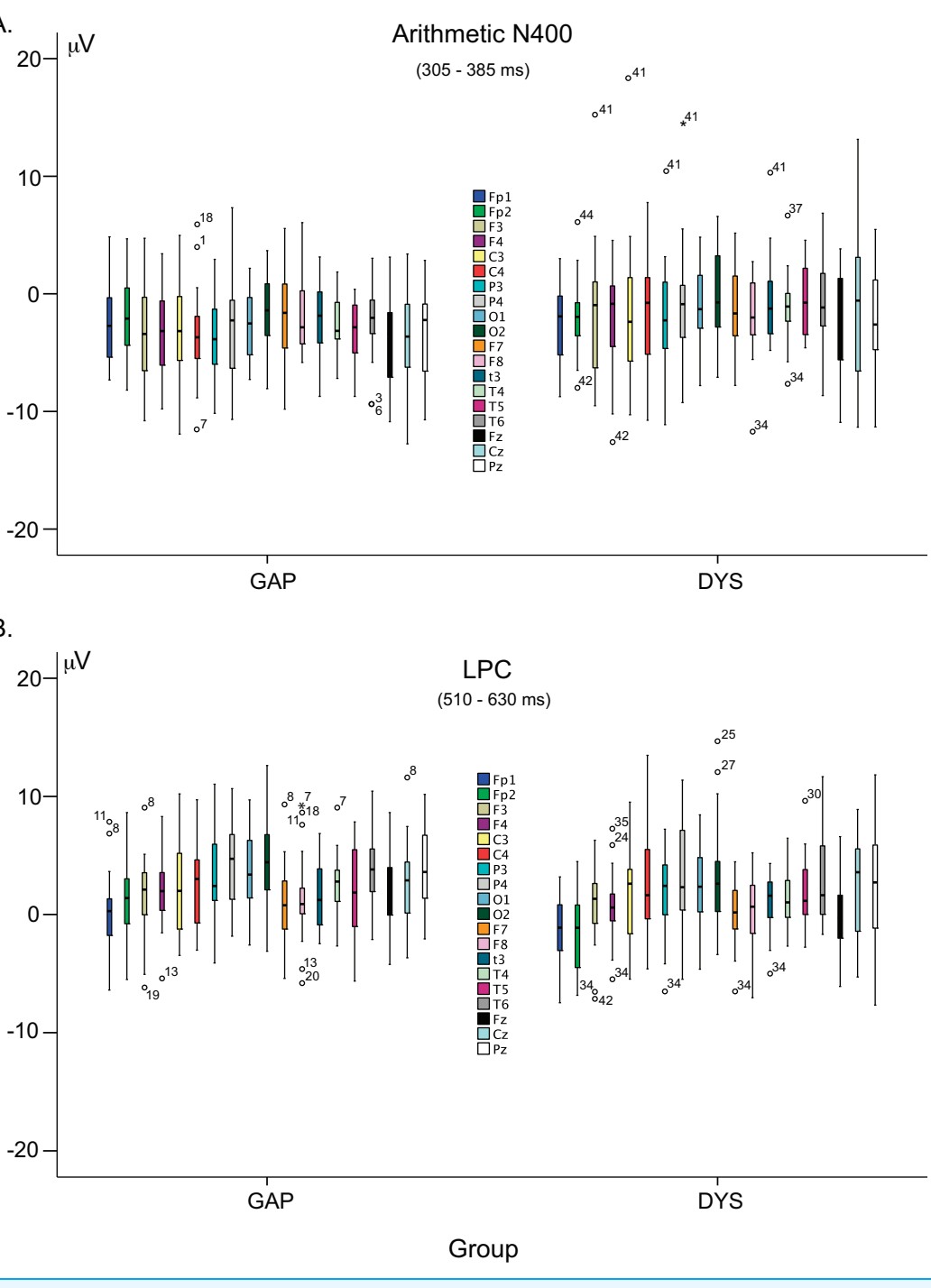

**Figure 8 Variability of arithmetic N400 and LPC effects.** (A) Box-and-whisker plots of both groups of children (GAP and DYS) using the amplitude values of the arithmetic N400 (305–385 ms) effect. (B) Box-and-whisker plots of both groups of children using the amplitude values of the LPC (510–630 ms) effect.

dyscalculia than in the children with GAP. For the ERP patterns, we hypothesised that the children with dyscalculia would display longer latencies and smaller arithmetic N400 and LPC effects than the children with GAP.

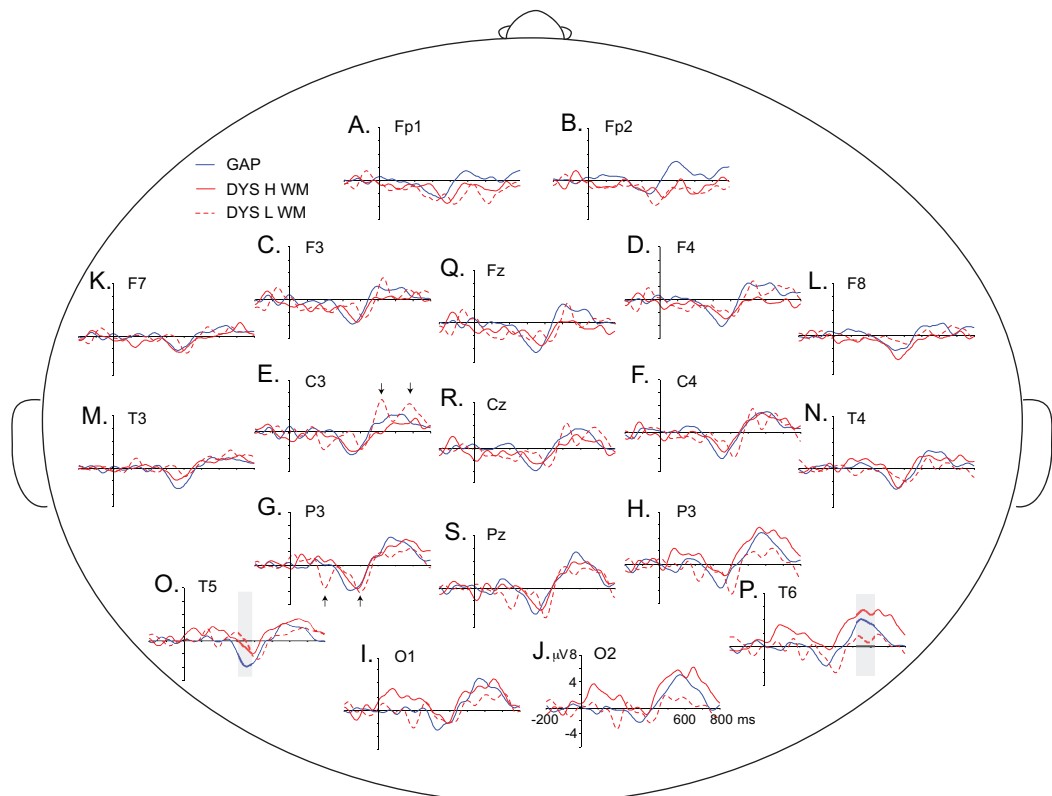

**Figure 9 Grand averages of the difference waves (i.e. incongruent minus congruent condition).** Blue solid lines represent the ERPs for the GAP group. Red solid lines represent the ERPs for the DYS group with high WM index scores and red dotted lines represent those for the DYS group with low WM index scores. Positive is plotted up. The arithmetic N400 effect and the LPC effect in the GAP group are marked with grey-shadow boxes. Black arrows indicate double-negative peaks (195 ms and 405 ms) and double-positive peaks (525 ms and 685 ms) in the DYS group with low WM scores at P3 and C3, but such effects can be observed over other electrode sites. Each letter represents an electrode. (A) Fp1. (B) Fp2. (C) F3. (D) F4. (E) C3. (F) C4. (G) P3. (H) P4. (I) O1. (J) O2. (K) F7. (L) F8. (M) T3. (N) T4. (O) T5. (P) T6. (Q) Fz. (R) Cz and (S) Pz.

## Behavioural differences between the DYS and GAP groups

Our behavioural results partially confirmed our hypothesis. We observed a significantly lower percentage of correct answers in the DYS group than in the GAP group. This result corroborates the findings of other behavioural studies (*Castro & Reigosa-Crespo, 2001*; *Geary, 1993*; *Geary, Bow-Thomas & Yao, 1992*; *Geary, Hoard & Hamson, 1999*; *Landerl, Bevan & Butterworth, 2004*). The poor performance of children with dyscalculia has been explained by their use of procedural strategies such as counting on, counting all, and decomposition, which are more prone to errors, instead of the long-term-memory retrieval strategies that are used by children with typical arithmetic abilities when facing one-digit addition problems (*Geary, 2004*). Unfortunately, in the present study, the strategies used were not systematically recorded for each child. This constitutes a limitation of the study because it precludes us from proving that the observed differences were attributable to the strategies used. On the other hand, there was no significant group difference in response times. This could be explained by the high dispersion in the data in both groups,

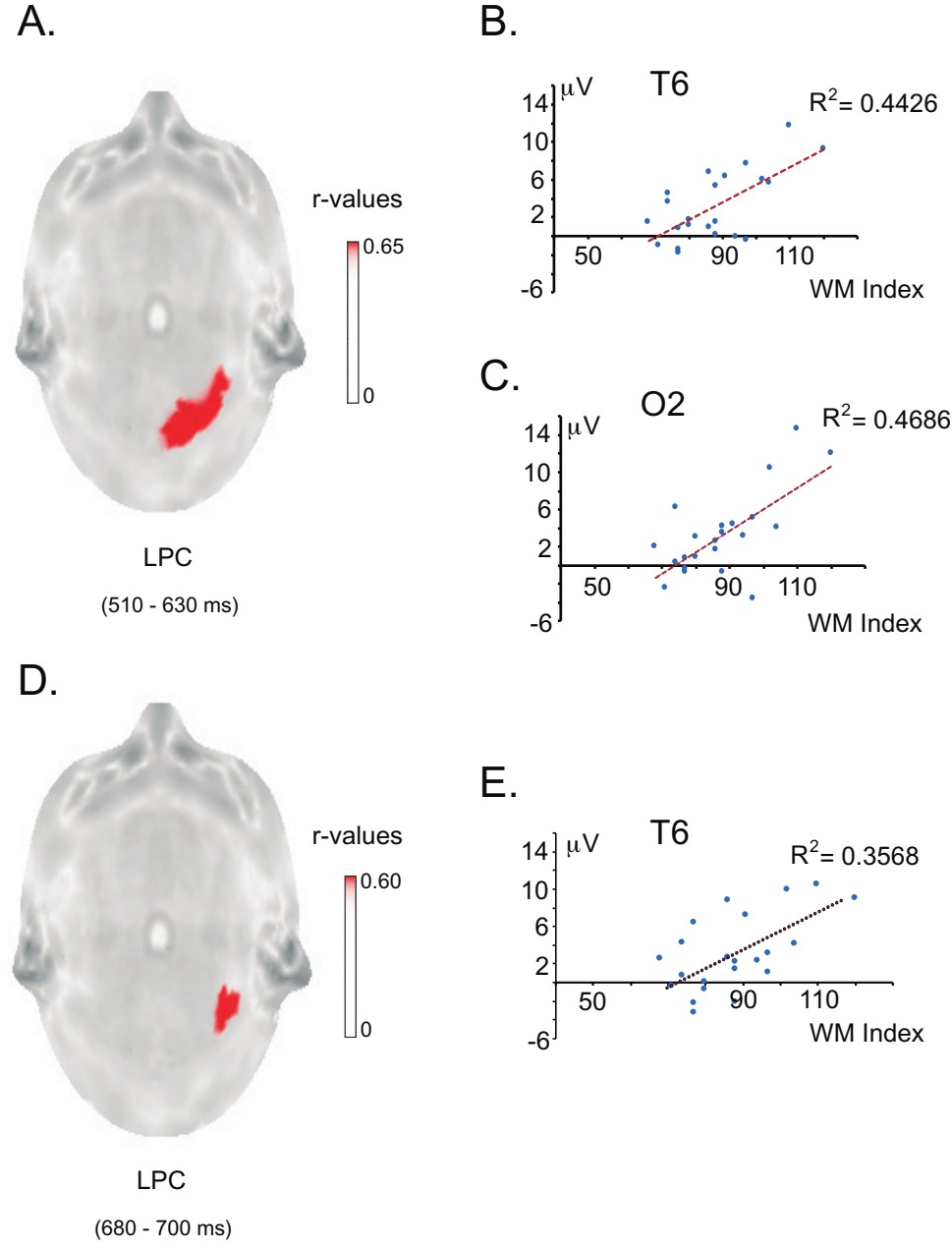

**Figure 10 Relationship between working memory and LPC effect in the DYS group.** (A) Statistical map of the correlations between the WM index and the ERP amplitude difference between conditions (incongruent minus congruent) at 510–630 ms (LPC effect) across electrode sites. The red spot represents the significant $r$ values ($p < 0.05$) over the T6 and O2 electrodes. (B) Ascending regression line showing that higher values of the working memory index ($X$ axis) are associated with greater LPC effects in the electrode T6 ($Y$ axis). (C) Ascending regression line showing that higher values of the working memory index ($X$ axis) are associated with greater LPC effect in the electrode O2 ($Y$ axis). (D) Statistical map of the correlations between the WM index and the ERP amplitude difference between conditions (incongruent minus congruent) at 680–700 ms (LPC effect for the DYS group) across electrode sites. The red spot represents the significant r values ($p < 0.05$) over the T6 electrode. (E) Ascending regression line showing that higher values of the working memory index ($X$ axis) are associated with greater LPC effects for the DYS group in the electrode T6 ($Y$ axis).

mainly in the DYS group (see Fig. 4). As expected, in the GAP group, older children showed shorter response times, perhaps because the automation of arithmetic facts tested herein is still developing in that age range. Interestingly, children with dyscalculia did not show this association of performance with age. This may be because, independently of age-related maturational process, children with dyscalculia experience problems in this automation process.

## ERP differences between the DYS and GAP groups
### N400 effect

Only the GAP group exhibited the arithmetic N400 effect (a higher amplitude for the incongruent condition than for the congruent condition). This effect was observed over the left temporo-parieto-occipital and right fronto-temporal regions and peaked earlier than 400 ms. The findings for the frontal region coincide with the topography observed in some studies in young adults (*Megías & Macizo, 2016*; *Prieto-Corona et al., 2010*) and the left posterior localisation coincides with those found in other studies (*Avancini, Galfano & Szűcs, 2014*; *Avancini, Soltész & Szűcs, 2015*; *Dickson & Federmeier, 2017*). This more-distributed effect in children corresponds with the findings reported by *Prieto-Corona et al. (2010)*, who observed that the N400 effect in children involves more cortical regions than that in adults to perform the same task, and by *Dong et al. (2007)*, who compared younger and older children during the performance of arithmetic verification tasks.

Only a few studies have assessed these effects in children, and the majority of them used different arithmetic operations, which activate different brain regions (*Zhou et al., 2011*). Another point of difference from these studies is that we obtained ERPs time-locked to the onset of the probe stimuli, whereas almost all studies obtained ERPs time-locked to the arithmetic problem or equation (*Van Beek et al., 2014*; *Xuan et al., 2007*). Only the study by *Xuan et al. (2007)* shows the same characteristics as ours; however that study observed the N400 effect over the vertex. One concern regarding ERP topography could be the use of non-parametric statistics because they are not commonly used. However, *Picton et al. (2000)* has argued that this is a better approach for ERP assessment than parametric analyses because it makes no assumptions about the distribution of the data, and is especially useful in the analysis of multichannel scalp distributions, as in our study. This is supported by *Megías & Macizo (2016)* who analysed their ERP data by using parametric and nonparametric statistical analysis and obtained similar findings with both methods, with the nonparametric permutations appearing to be more sensitive to differences.

Since we are using an arithmetic verification task to evoke the ERPs, it is important to determine which processes could manifest in it. Among the four processes involved in the arithmetic verification task proposed by *Avancini, Soltész & Szűcs (2015)*, two were controlled in our task: (1) the number of congruent and incongruent probes was equal, so violations of strategic expectations should not have manifested as ERP effects; and (2) precisely the same probe stimuli were used for both conditions, so the physical characteristics of the visual stimuli would not have affected the ERPs. The other two effects are the magnitude effect and the violation of the operands' semantic constraints when an

incongruent probe is shown. Although all the incongruent probes were two units away from the correct solution in our paradigm, a magnitude effect may have been present; therefore, the priming effect and the magnitude effect could be mixed. A stronger left posterior effect related to distance was observed by *Avancini, Galfano & Szűcs (2014)*, consistent with the studies indicating the association of this area with the verbal code according to the triple-code model (*Dehaene & Cohen, 1997*). In our study, the GAP group showed a higher N400 effect than the DYS group precisely in the left posterior temporal area (Fig. 8).

In contrast, children with dyscalculia showed no significant arithmetic N400 effect, and when their ERPs were compared to those of the controls, significant differences were observed over the left posterior temporal region. This finding is consistent with those of studies reporting a smaller arithmetic N400 effect in adults or teenagers with dyscalculia compared to age-matched controls (*Núñez-Peña & Suárez-Pellicioni, 2012*; *Soltész et al., 2007*). The lack of a significant N400 effect in children with dyscalculia could be explained as a failure to process congruent results. In these children, any probe (congruent or incongruent) is perceived as a mismatch with what is stored in the arithmetic lexicon (in Fig. 6A negative deflection is elicited in both conditions). Thus, they must revert to conducting the arithmetic calculation. The group differences in the left temporal region may reflect the fact that simple addition problems activate phonological processes, as has been described for multiplication problems (*Zhou et al., 2009*).

### LPC effect

The LPC effect was displayed in both groups, but with different latencies and topographies. The DYS group showed a delayed LPC effect of shorter duration. Since the LPC effect is modulated by the expectation or plausibility of the solution and children with dyscalculia had lower arithmetic abilities, we expected a smaller LPC effect in the DYS group than in the GAP group. Our results support this hypothesis because a significantly lower amplitude of the LPC effect was observed in the DYS group in the right frontopolar region. Like other studies (*Iguchi & Hashimoto, 2000*; *Núñez-Peña, Gracia-Bafalluy & Tubau, 2011*; *Núñez-Peña & Honrubia-Serrano, 2004*; *Núñez-Peña & Suárez-Pellicioni, 2012, 2015*; *Szűcs & Soltész, 2010*), we observed that the LPC effect is greater in individuals with better performance, and that this difference was located in the right frontal region. *Meiri et al. (2012)*, who used functional near-infrared spectroscopy, observed that the right frontal region is activated during simple additions, and this region is believed to be responsible for holistic arithmetic processing (*Dehaene et al., 2003*; *El Yagoubi, Lemaire & Besson, 2003*). This suggests that children in the GAP group perform a greater re-evaluation of incorrectness when the proposed result was incongruent than when it was congruent, while children with dyscalculia, perhaps due to the lack of arithmetic knowledge, re-evaluated almost all the results without distinction between congruent and incongruent conditions.

Differences in topography were also observed between groups: The GAP group showed the LPC effect in the expected right posterior location, while the DYS group exhibited this effect in the left posterior region (see Fig. 5). The right lateralisation of the LPC effect

in children with GAP is consistent with the more deliberative and prolonged role of the right hemisphere during probe evaluation, which has been found in adults during a multiplication verification task (*Dickson & Federmeier, 2017*). According to these authors, after an initial period of evaluation of the provided response (probe), the left hemisphere classifies it as correct or incorrect and no longer performs follow-up evaluations, while the right hemisphere engages in a deliberate assessment of the additional features of the probe, perhaps using spatial skills, to provide an evaluation that is less categorical. It is therefore possible that children with dyscalculia intentionally search for the correct answer from their long-term memory (left hemisphere), but failing to find the answer, they then perform the arithmetic calculation. Although the topography recorded from the scalp does not necessarily indicate the generators' location, different topographies indicate the presence of distinct generators (*Nunez & Srinivasan, 2006*). Our results may suggest that the left lateralisation of the LPC effect observed in children with dyscalculia is a compensatory phenomenon to obtain the correct answer.

## Heterogeneity within the DYS group

In contrast to our expectations, we found few differences between groups in the arithmetic N400 and LPC effects. The heterogeneity in the children's behaviour (Figs. 1 and 4), which was enhanced in the WM behavioural scores (Fig. 1B), and ERP patterns (Fig. 7) of the DYS group, could explain this finding. Two main hypotheses have been proposed to explain atypical brain functioning that is reflected as neurobiological disorders of cognitive processing (*Silver et al., 2008*) that underlie learning disorders (*Landerl et al., 2009*). In addition to the *domain-specific hypothesis*, which refers to abilities specifically related to mathematical competencies, the *common-deficit hypothesis* postulates that certain processing patterns are common to all children with learning disorders. Supporting this hypothesis, *Swanson (1987)* proposed that children with learning disorders experience failures in executive functioning mechanisms, which also points to WM deficits as essential problems (*Berninger, 2008*; *Swanson, 2015*; *Swanson & Siegel, 2001*). In children with arithmetic disabilities, WM has been frequently reported to play an essential role in the arithmetic domain (*Swanson, 2015*). In our study, once children had performed the addition operation, they had to store the result in WM until the probe digit appeared (1,500 ms later) to perform the response verification process and finally provide an answer. Therefore, the arithmetic verification task that we used is particularly efficient for highlighting WM problems.

## WM and dyscalculia

Consistent with our hypothesis, the children with dyscalculia showed a lower WM index than those in the GAP group. This finding aligns with previous studies where WM was found to predict learning arithmetic (*Meyer et al., 2010*; *Vanbinst & De Smedt, 2016*), as well as a study by *Mammarella et al. (2017)*, which reported that children with dyscalculia had low scores for WM. Since the arithmetic N400 effect reflects a facilitation for the probe stimulus that matches the correct answer, it may be the case that the absence of this

effect is associated with poor WM. Keeping the information of the addition in WM, as children with GAP likely do, facilitates recognition or rejection of the proposed result.

However, it is important to emphasise that the WM performance in the DYS group was not homogeneous. And while exploring the relationship between WM and arithmetic processing in the DYS group, we discovered that children with higher WM index scores showed a greater amplitude of the LPC effect in the right posterior region. This region coincides with the LPC topography observed in previous studies (*Niedeggen & Rösler, 1999*; *Núñez-Peña & Escera, 2007*; *Núñez-Peña & Honrubia-Serrano, 2004*) and in our control participants.

This relationship between WM and the LPC effect was elucidated in the present study and contributes to the understanding of dyscalculia in children. For a more thorough exploration of the WM effect in children with dyscalculia, children in the DYS group were classified into two groups (average and lower-than-average) according to their WM index. Visual inspection of ERP patterns from these two groups showed that the children with dyscalculia and an average WM index had a similar ERP pattern to that in the children with GAP, while the children with dyscalculia and a lower-than-average WM index showed an atypical ERP pattern (Fig. 9). Visual inspection of the ERPs suggests that this atypical pattern consisted of two negative peaks (at 195 ms and 405 ms) over the parieto-occipital and centro-parieto-temporal regions and two positive peaks (at 525 ms and 685 ms) over the parietal regions. The two negativities could correspond to the N200 and arithmetic N400 effects, while the two positivities may correspond to the two LPC effects. The N200 effect might be interpreted as evidence that children with dyscalculia and poor WM engaged additional attentional resources (*Xuan et al., 2007*). However, this N200 effect had a posterior topography; which may instead reflect a strong early sensory attention (*Schmajuk et al., 2006*) before the comparison between the probe stimulus and the sum result, which produces an arithmetic N400 effect. Later, the children probably re-evaluated the arithmetic error (*Núñez-Peña & Suárez-Pellicioni, 2012*) twice.

It is noteworthy that the categorisation of the ERP patterns of the DYS group into two subgroups was based on visual inspection. Ideally, we would have compared the ERPs of the children with dyscalculia with poor WM and typical WM statistically, but the sample sizes of these two subgroups were too small. It would be useful if future studies could conduct these statistical comparisons to help clarify whether the atypical ERP pattern that we observed is reliably association with dyscalculia, poor WM, or both difficulties combined.

## CONCLUSIONS

Children with dyscalculia did not show the arithmetic N400 effect found in children with typical development during an arithmetic verification task; however, both groups showed an LPC effect. The great heterogeneity within the group of children with dyscalculia precluded a robust LPC effect in these children; however, the higher the WM deficits were, the lower the LPC effect was in the right posterior region. In children with dyscalculia and WM deficits, an atypical ERP pattern (i.e. N200, N400 and two LPC effects) was

evinced. Therefore, future studies of both WM and ERPs in children with dyscalculia must be mindful of the heterogeneous nature of dyscalculia at both the level of behaviour and the brain function.

## ACKNOWLEDGEMENTS

The authors are grateful for the cooperation of the children and parents who participated in this study. The authors also acknowledge the administrative support provided by Bertha Esquivel, Iva´n Negrete, Leonor Casanova, Lourdes Lara, and Teresa Alvarez, and the technical assistance provided by Benito Martínez-Briones, Daniel Villareal, Enrique Cabral, Héctor Belmont, Lucero Albarrán-Cárdenas, María del Carmen Rodríguez, Maria do Carmo Carvalho, María Elena Juárez, Marisa Oar, Mauricio Cervantes-Romero, Milene Roca Stappung, Minerva Berenice Rojas, Roberto Riveroll, Saulo Hernández, and Sergio Sánchez-Moguel. We also thank Angelica Acosta for her comments on the manuscript.

### Funding

This work was supported by the Programa de Apoyo a Proyectos de Investigación e Innovación Tecnológica (IN204613, IN205520, and IN207520) and the Consejo Nacional de Ciencia y Tecnología (CONACYT; CB-2015-01-251309). Sonia Y Cárdenas is a beneficiary of the CONACYT scholarship (No. 336175). The funders had no role in study design, data collection and analysis, decision to publish, or preparation of the manuscript.

### Grant Disclosures

The following grant information was disclosed by the authors:
Programa de Apoyo a Proyectos de Investigación e Innovación Tecnológica: IN204613, IN205520 and IN207520.
Consejo Nacional de Ciencia y Tecnología (CONACYT): CB-2015-01-251309.
CONACYT scholarship: 336175.

### Competing Interests

The authors declare that they have no competing interests.

### Author Contributions

- Sonia Y. Cárdenas conceived and designed the experiments, performed the experiments, analyzed the data, prepared figures and/or tables, authored or reviewed drafts of the paper, and approved the final draft.
- Juan Silva-Pereyra conceived and designed the experiments, performed the experiments, analyzed the data, prepared figures and/or tables, authored or reviewed drafts of the paper, and approved the final draft.
- Belén Prieto-Corona conceived and designed the experiments, performed the experiments, analyzed the data, prepared figures and/or tables, authored or reviewed drafts of the paper, and approved the final draft.

- Susana A. Castro-Chavira performed the experiments, prepared figures and/or tables, authored or reviewed drafts of the paper, and approved the final draft.
- Thalía Fernández conceived and designed the experiments, performed the experiments, analyzed the data, prepared figures and/or tables, authored or reviewed drafts of the paper, and approved the final draft.

## Human Ethics

The following information was supplied relating to ethical approvals (i.e. approving body and any reference numbers):

The Bioethics Committee of the Instituto de Neurobiología at the Universidad Nacional Autónoma de México (UNAM) approved the experimental protocol (INEU/SA/CB/145).

## Data Availability

Raw data is available at Figshare:

Cárdenas-Sánchez, Sonia Y; Fernandez, Thalia; Silva-Pereyra, Juan; Prieto-Corona, Belén (2020): Arithmetic Processing in Children with Dyscalculia. An Event-Related Potentials Study. figshare. Dataset. DOI 10.6084/m9.figshare.9739052.v1.

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
