# Peer review of "Arithmetic processing in children with dyscalculia: an event-related potential study"

_PeerJ, doi:10.7717/peerj.10489_

## Round 0.1 · original submission · Major Revisions

Dear authors,

Thank you for submitting your manuscript "Arithmetic processing in children with dyscalculia. An event-related potentials study" for peer review in PeerJ.

Your manuscript has been reviewed by four reviewers, including myself as the fourth reviewer. I concur with the points and suggestions offered by the first three reviewers, and suggest that you respond to all suggestions offered by all reviewers - in the manuscript itself or in a response letter - before resubmitting to PeerJ for a second round of reviews. It would be helpful if you could also indicate what changes you have made to the manuscript, and where those changes can be found.

As you can see from my own comments (see below under Reviewer 4), the manuscript was not easy to follow in places (particularly the Introduction and Discussion) which makes the validity and reliability of the study hard to assess with confidence. Should these major revisions be completed successfully, I anticipate the manuscript will require further (potentially multiple) revisions once the experimental methods and outcomes become clearer. The authors may wish to take this into consideration.

We all hope that you find their comments and suggestions constructive in further improving your work.

Your sincerely,

Genevieve McArthur

REVIEWER 4 (also Academic Editor)

Basic reporting

1. I agree with the second reviewer that the working memory component of this manuscript is not strong. Working memory is touched upon (very) briefly in the Introduction, and then a leap in logic is made in the Discussion that the findings somehow reflect working memory deficits in participants. This link between working memory and the ERPs is not clearly justified in the manuscript, and the reader is left wondering how or why such a leap could/would be made - particularly when working memory is not explicitly tested within the study. Unless there is a good reason to include the working memory component in the study, I would suggest removing it from the manuscript. If the authors believe that it should be considered, then the study needs to demonstrate, somehow, that the dyscalculia sample actually had poor working memory. Further, given the heterogeneity that characterises dyscalculia, it is unlikely that every participant in the dyscalculia group has poor working memory as measured by a non-numerical working memory test. Hence, the study would need to show that the children with dyscalculia and poor working memory are the same children with atypical ERPs.
2. This leads to the point of heterogeneity of dyscalculia, which the second reviewer also mentioned. dyscalculia is a heterogeneous condition: not all children have the same cognitive deficits. Over the years, much time has been wasted examining ERPs in children with developmental disorders without reporting or discussing the individual differences. This paper should show and consider discuss the individual differences within the dyscalculia and control groups so that readers can understand how widespread the atypical ERPs might be in the dyscalculia population. We also need a better understanding of the variance in the ERPs in typically developing children (i.e., we need graphs that compare individuals' waveforms and ERP scores in both groups). Based on previous research with other developmental disorders, it is inevitable that some dyscalculia children will show atypical ERPs and some will not. This is an important thing to explore and address and discuss.
3. The Introduction needs more work. In many places, it reads like a list of statements that do not follow logically from each other. For example, lines 66 to 76 provides a definition of dyscalculia. This is immediately followed by a statement that children with dyscalculia have working memory problems (lines 77-80), which is in turn followed by a brief statement that children with dyscalculia have been found to have atypical electrophysiological patterns (81-89). The lack of segues between such statements obscure the flow of logic of the argument and hence the point of raising those issues. This problem is particularly apparent at the end of the Introduction where the manuscript simply states the objectives of the studies without justification of why those objectives are being addressed, other than "this is the first study" to do so (this is not a sufficient reason from a theoretical point of view), and then lists the hypotheses without explaining why those hypotheses were made.
4. These issues contribute a major limitation of the Introduction and Discussion, which is an apparent lack of theory behind the study. The reader is not provided with a clear idea of why the study is being done, and what the outcomes will tell us one way or another in terms of our theoretical understanding of dyscalculia, or what atypical N400 and LPCs actually mean in terms of cognition. This could be improved, in part, by a more complete discussion of what the N400 and LPC has been used in previous research, what these peaks appear to represent (including reference to language and reading research), and why it is important to measure these peaks in children with dyscalculia (e.g., what can these peaks tell us about the mechanisms involved in dyscalculia?).
5. The use of grammar and punctuation throughout the manuscript needs careful review. In general, the writing is fine. However, there are numerous points where incorrect punctuation is used (e.g., the use of ; instead of .) and the grammar is incorrect (e.g., "at least 1 times the minimum wage"; "addition fact"). In addition, it is very hard to tell where paragraphs begin and end. Please use indents or insert blank lines to make paragraphs clearer.
6. Avoid the use of abbreviations where-ever possible. It puts extra cognitive load on the reader unnecessarily, impairing easy understanding.
Experimental design

7. The experiment tested children on the WISC-IV (line 173) Please include the data for each subtest (and overall standard score) for each group in Table 1 so the reader gets a clearer idea of the cognitive profile of each group.
8. How did professional educators confirm the dyscalculia diagnosis (line 187)? And why was this necessary given the current study tested the children for dyscalculia according to set criteria.
9. Under ERP processing (line 245), it is said that baseline correction was done after epoch rejection. But usually epochs need to be baseline corrected before they are rejected - particularly if criteria regarding amplitude are used to reject epochs (in this case, over 100 mV). Please justify.
10. The number of epochs that were included to create ERPs for each condition for each child need to be reported so the reader can assess how reliable the ERP data may be (line 248).
11. I found the Statistical Analysis section difficult to follow. I suggest this section is merged with the Results section. Specifically, under Results, explain that the analysis involved a number of (X) steps. Then, for each step, explain (1) what was done and (2) justify why that step or particular analysis was done (i.e., why was this analysis selected rather than another?). This needs some careful work so the reader can better assess the reliability and validity of the findings.
12. On lines 292-293, it is reported that children used their fingers during the experiment. It was not stated in the Methods that such observations would be made. Neither was it stated how such observations were measured. This information currently looks like an anecdotal afterthought that has been inserted to justify an interpretation in the Discussion. As such, this interpretation is not convincing as it stands.
Validity of findings

13. Line 333 refers to a priming effect. This seems odd to me because what is described is a congruency effect rather than priming effect.
14. The manuscript provides a very cursory list of limitations. The limitations are not described in detail, and the potential impact of those limitations on the outcomes of the study are not explained clearly. The limitations of this study must be considered in much more detail to allow the reader to assess its potential validity and reliability.

·

Basic reporting

The paper reports an experiment comparing children with dyscalculia and children with good academic performance on an arithmetic verification task.

The paper is well written, structured and easy to read. The figures are adequate and the authors have done a good job supplying the raw data. In fact, I was able to re-analyse their data (using a slightly different method) and was able to confirm the results presented in the paper.

In terms of the figures, I *personally* do not like bar charts and would instead recommend an alternative. For example, representing the mean with a dot and adding error bars. In terms of the error bars, the nature of the error bars (i.e., SD, SEs, or CIs) should be included in the figure caption for figure 2.

Experimental design

no comment. Well done.

Validity of the findings

I was able to confirm the results presented in the paper through a re-analysis of the supplied data.

It is unfortunate that the sample size is a little low for a between-groups comparison. However, I completely understand the difficulty associated with recruiting children with dyscalculia, and therefore in this context I would consider the sample size as adequate.

·

Basic reporting

1) Literature references
- In the manuscript, most of the literature related to the main topic has been covered and there is a good effort by the author. Considering the conclusions and speculations made by the author in the discussion, references to the working memory deficits in dyscalculia are missing; this part should be expanded in the introduction.
2) Language
- Language is appropriate, although in some parts it could be improved. For example, inline 81 and 83, the author should report directly reference and findings and avoid “various studies have claimed that”, “have shown that”
3) Introduction
- I would like to suggest to the author to rewrite the definition of dyscalculia and to refer to the latest research on the subject by underlying the complexity and the heterogeneity of the disorder.
- Lines 81-82. The connection to IPS if not contextualized adds nothing to the text. The author should add some references to the role of the IPS, which can be relevant for explaining their explanation of a deficit in working memory.
- Lines 89-91. They are not clear and they need to be rephrased and expanded.
- Lines 114-115, 118. The author is repeating the same concepts mentioned before.
- The sentences 138-139 and 155-156 are in opposition and confuse the reader.
4) Figures
- I would suggest adding a plot with the difference wave (incongruent minus congruent). For the reader, it would be easier to see the difference between the N400. Further, it is more clear to have a 2D scalp topography, not a 3D.

Experimental design

Design
- The design and the method used are clearly explained in detail.
Some minor comments:
How many children did the author discarded from the analysis for a noisy signal?
Rereference of the EEG signal is missing.
Lines 244-245. Not clear

Validity of the findings

Validity of findings
1) Statistical analysis is computed correctly and are clearly explained. They are in line with recent publications.
Minor comment
- in the results section, it is not clear on which electrodes the statistics are reported.
- Finger counting should have been assessed or observed systematically in order to be reported
2) Discussion
- the author should try to explain why there is no difference between the two groups in RT.
- Lines 353-359: I would suggest the author to try to find other references in support of the proposed explanation of a WM deficit. In the dyscalculia literature, few authors highlights the deficit of WM (Szucs et al., 2013 2014). The common deficit hypothesis seems too general.
Lines 731-734: I find this part too general, without references and a clear explanation about the link between N400 and working memory.
- Lines 375-377: not clear

Additional comments

The research question is interesting and the current study can contribute to the existing literature. Overall, the article is well written and most of the relevant literature is covered. However, the introduction and the discussion could be improved. Importantly, the authors try to explain the difference between the two groups in this task as a difference in the ability to maintain information. I mainly ask them to contextualize it, explain it better and highlight the link between the N400 component and the working memory.

·

Basic reporting

In general, the introduction section contains a solid background for the proposed study grounded in the literature. However, there are quite some errors in the use of the English language, not only in the introduction, but throughout the manuscript.

Tables and Figures are high quality and well described.

Raw data are not supplied. Only averages per subject per condition are supplied. What is missing is the raw EEG data. Also, the behavioral data are missing.

Some parts of the introduction can be clarified:
- Line 112-113: what ages are included in the middle-ages and young people in this research? I’m asking since I think there is an ‘optimal’ age, and whether you see better or worse performance in the older age-group depends on the ages included.
- Line 138: ‘the topographical distribution was different’. As an explanation, the authors write that the amplitude of the component differs between groups. This is something different than the topographical distribution.
- At the end of the introduction, the authors state the hypotheses. However, these hypotheses (especially hypothesis b) are not linked with the rest of the introduction. Can the authors make explicit why they expect a lower amplitude/ higher latency in the children with dyscalculia?

Experimental design

The research question is well defined and relevant. However, the knowledge gap could be made more explicit in the introduction.
I have some comments and questions on the methods and results sections:
1. I miss a justification of the number of participants that were included in the study. Although the authors mention in the discussion that the sample size was small, I doubt whether the sample is sufficient. Although there are no clear cut rules on the minimum number of participants, I think it is generally acknowledged that one needs at least 20 participants per group. Are the authors able to show either a power analysis or a justification for the sample sizes? I know there are studies with less participants than 20 (including my own), but in the light of the recent developments in the area of reproducibility, I think it is crucial to have a justification for the small sample.
2. How many children were excluded from the study? The participants were selected from a sample of 152 children, but it is not clear how they were selected. I’m concerned about this especially given the last paragraph of the discussion, in which the authors write that many students with dyscalculia did not meet the criteria, including the criterion of enough correctly answered trials. As such, I am wondering as to whether the sample of children with dyscalculia is representative of the population (or the larger sample) of children with dyscalculia.
3. In line 224 the authors write that the participants were instructed to answer as fast as possible. However, they could only respond after 3000 ms, when the answers became visible. Once the participants knew that they had 3000 ms to respond, how could the authors check whether the children indeed tried to come to a solution as soon as possible? The observed use of finger counting suggests that the children did not answer as fast as possible
4. The authors write in line 247-248 that the average number of segments per condition was approximately equal. How many segments were included? And more importantly, how many segments were included in the GAP group and in the DYS group? In the results section, the authors report a larger N400 effect in the GAP group. However, this could be due to the inclusion of more trials, and thus less noisy data leading to larger amplitudes. Therefore, it is important that a similar amount of trials are used to generate the ERPs in both groups and both conditions.
5. In line 267 and 268, the authors write that they used separate analyses for each group. Can the authors explain why they did this instead of looking at an interaction between Condition and Group?
6. In Figure 2, the correct answers in the DYS group are around 60%. Did the authors test whether this performance was above chance? Otherwise, they might not be measuring what they intend to measure in the ERPs.
7. The authors write about a difference in OMs between the groups. However, no descriptives on the OMs are provided.
Minor comments in the methods sections:
a. Line 182: numeric management is not a common term. Can the authors explain what they mean with this term?
b. Par 2.3 ‘procedure’ actually contains specifics about the task. I think that should be a separate header
c. Line 230: children were instructed to avoid blinking. Were there moments in between the trials in which the participants were allowed to blink?
d. Line 309-310: which window was chosen to test this difference in N400 effect?

Validity of the findings

With regard to the standards of the journal, not all underlying data have been provided. As stated before, the raw EEG data as well as the behavioral data are missing. Moreover, an analysis script and the results of the permutation test would be helpful, especially since permutation tests will always return slightly different results when being re-run.
In the discussion, I have difficulty following the authors’ line of thought. The subdivision in paragraphs is not clear:
1. The authors write that poor performance of children with dyscalculia can be explained by the use of procedures instead of retrieval (line 334 and further). Whereas this might be the case, the use of finger counting in children in the GAP group suggests that these children do not use retrieval strategies either. The authors also acknowledge this, but this undermines the likelihood of their explanation.
2. From line 353 the authors write that there are 2 main hypotheses to explain learning disorders, however, they only mention one of them. Then they continue with ‘on the other hand’, however, it is not clear what ‘on the one hand’ is here? This makes the text difficult to follow between line 353 and 364.
3. I don’t understand what is meant in line 370 until 374.
4. In line 387 the authors write that they don’t find differences. I think they mean that they do not find differences in amplitude.
5. Line 392-394 needs to be elaborated. What does this mean (functionally) that the LPC does not (yet) have a parietal distribution?

Additional comments

In this manuscript, the authors study arithmetic processing in children with and without dyscalculia using ERPs. This is an interesting topic given the problems children with dyscalculia face in basic arithmetic. However, I do have some concerns about the manuscript. The most crucial point is the sample. Can the authors justify their small sample? And can they justify the selection of their sample (see 'Experimental design', point 2)

---

## Round 0.2 · Minor Revisions

Dear Dr Cardenas and team,

Thank you for submitting a revised version of your manuscript "Arithmetic processing in children with dyscalculia. An event-related potentials study" to PeerJ for a second stage of reviews.

I sent your revised manuscript to the two reviewers who made the most suggestions on the first version of the manuscript. They both appreciate the effort that you have put into revising the manuscript according to the suggestions, and agree that both the study and manuscript are significantly improved. After their second review of the manuscript, they both offer minor suggestions to improve the manuscript further. I suggest that you follow the suggestions of both reviewers closely, so we can continue to move forward on this piece of work.

A recurring suggestion from reviewers in both review rounds was the need to improve the clarity of language and organisation of the manuscript. While the revised version of the manuscript has improved in this respect, it is still difficult to follow in numerous places. I understand that you had your manuscript checked for English language by a professional company prior to the original submission. Unfortunately, they do not seem to have served you as well as you needed. I suggest that after you have made your revisions, you have the next version of the manuscript proof-read by a few English-speaking scientists in our field of research, before you resubmit. Or perhaps find a new proof-reading company that will be a higher-quality job. This manuscript cannot be published unless the language and logical flow of the written arguments reach the appropriate standard for scientific writing.

If you get really stuck, perhaps you could contact PeerJ to explain the situation and ask if they have any advice or recommendations for support in this respect.

I hope you are staying safe and healthy during this difficult time,

Kind regards,

Genevieve

·

Basic reporting

Firstly, I would like to thank the authors for the excellent work they did for the revision. Improvements are clearly visible and the great effort is evident. Particularly, following the reviews, they deepened the link between WM and mathematical skills, reporting very relevant references. This allowed them to clarify their hypothesis and their method of investigating this relationship with the current study in DD. Additionally, in the methodological section, relevant details regarding the ERP analysis have been reported. Here are some minor comments.

Experimental design

-To improve the section regarding the stimuli, I suggest explaining additions are generated. Digits used from 1-9. How much the incorrect probe differs from the correct one? Was it constant? What about carry-nocarry (eg. 9+4 or 5+3)? Was the same number of carry or no carry trials in the two groups?
- It is good that nonparametric analysis on ERPs was computed. However, these findings need to be shown that they are compatible with the previous literature in terms of regions.
- which are the tests of WM in the WISC the authors correlating with the ERP measures? In the WISC there are 3 WM subtests. The average?

Validity of the findings

- It is important to link ERP finding with the previous findings and compare them to show reliability.
- The differentiation between low and high WM in DD children requires additional attention. Making assumptions regarding low and high WM in DD group without statistical results can be risky. It can be certainly commented on and emphasized for future studies. Unfortunately, the number of DD children does not allow further analysis.

·

Basic reporting

Although the language has improved in general, I still have difficulty understanding some parts of the text:
- The introductory part of the abstract is not clearly structured, i.e. sentences do not follow each other logically.
- Also, the last sentence of the discussion part of the abstract is unclear without having read the full manuscript.
- Line 95 starts with ‘therefore’, but it is not clear what this term refers to
- Line 100-102 does not make sense. It states that individuals with good ability perform better than those with poorer performance, which is circular reasoning
- In line 115-116, the authors claim that a child need to maintain a result in WM. The result actually needs to be maintained in verbal short term memory. Earlier in the introduction, the authors mention that children with dyscalculia mainly have difficulty in visuo-spatial STM and the central executive. This doesn’t align with each other
- Line 145: ….but no significant differences……. Should be: but not significant amplitude differences
- In line 171-172, the authors refer to ‘the ERP components’. I assume that this refers to the N400 and LPC, but this should be mentioned.
- In the paragraph starting at line 170, the authors talk about the relation of verbal WM with math achievement. They argue that children with dyscalculia will have different ERP patterns based on the relation between verbal wm and math achievement. However, earlier (line 86-94) they claim that it is mainly visuo-spatial wm and central executive that children with dyscalculia have problems with. As such, this does not align with each other. Children with dyscalculia have also shown to display problems in verbal wm, but that should then be added in the paragraph starting at line 86.
- Line 176 – 179 would benefit from re-phrasing: 1st aim and second aim

Tables and Figures are generally clear. However, Figure 4 and 8 are thresholded. As such, they only show the locations of the significant t-values. In the original manuscript, there was no thresholding, which I like better, because it gives more insight into the underlying data.
I don’t see much added value of Figure 9. I think scatterplots of the correlations at T6 and O2 would give better insight into the data.

For the previous version, I made the remark that the raw data are not supplied. I had misunderstood the data provided by the authors and missed the behavioral data-file which was included. The data that are provided do meet the criteria for reproducibility.

Experimental design

The research question is well defined and relevant. It is very beneficial that the authors included more participants. I still have some comments and questions on the methods and results sections:
In the reply to my 2nd and remark, the authors write how many children were excluded and why. However, this should also be included in the manuscript. The same holds for my comment on ‘participants were not allowed to blink’.

The results-section has become much clearer. However, I think it would be better to re-order a part of this section. I think line 356-366 should be placed directly after line 344.
In line 3356-357, the authors talk about ‘two statistical analyses for independent samples’. Were permutation tests used here as well?
Another question about this analysis is why the authors did not test differences between the groups in the time window in which they found the LPC for the dys-group?
In line 371 the authors report a correlation with the LPC-effect. They should mention that this is the LPC-window for the GAP-group (or mention a time window), since they have two LPC windows (1 for GAP and one for DYS).

Validity of the findings

The discussion is clearer than in the first version of the manuscript. However, I still have some questions:
In line 390 the authors refer to the use of more procedural strategies instead of retrieval practice in children with dyscalculia. What is missing in their reasoning is that procedural strategies are more prone to errors than fact retrieval.
In line 395 the authors write that there were no differences in response times. However, I think this is a weak argument, because the arithmetic problem had already been displayed 3 seconds ago once the participants could answer.
The authors write that there was much dispersion in the RT data. This might be due to the age range of the participants, since automatization of the arithmetic facts tested here still develops in that age range. One could try to see whether there is a relation between RT and age to elucidate this potential relationship.
Line 414-416 are unclear to me.
In line 428-431 the authors write that there were no significant differences between the groups, whereas I think the results in line 363-364 show a difference in Fp2? If that is the case, than the authors should also be careful with their interpretation, because the difference between the groups is not in the right posterior location, but in the frontal regions.
With regard to line 460-462 I still think it is unlikely that children only start to calculate after the probe is presented, since they were instructed to answer as quickly as possible.
In line 490 and further, the authors write that the a-typical pattern in children with low WM may be consist of an N200 and an N400, the latter of which is due to matching the probe to the result to the equation. However, earlier in the discussion, they explain the absence of the N400-effect in the DYS-group as not processing the congruent result. So, it seems that the finding of an N400-effect in the low WM group is contrary to the expectations? And the explanations seems to contradict each other.
In line 512 the authors mention the lower LPC amplitude. It should be added that this holds for the LPC amplitude in the right temporal regions.

Additional comments

In this manuscript, the authors study arithmetic processing in children with and without dyscalculia using ERPs. This is an interesting topic given the problems children with dyscalculia face in basic arithmetic.
The manuscript has majorly improved compared to the previous version. I definitely think it is an interesting addition to the existing literature. However, I did still find some missing information and sections that are difficult to follow, so I would like you to incorporate some additional changes.

---

## Round 0.3 · Minor Revisions

Dear Sonia and colleagues,

Thank you for your latest version of your manuscript, which addressed most of the issue outlined by Reviewers 2 and 3.

I think we are at the stage where the reviewers have provided as much support as we can ask of them, so it is time for us to work to together to get the manuscript to a publishable level. As I mention in my previous communication to you, a recurring suggestion by reviewers is the need to improve the clarity of language and organisation of the manuscript. I note in your response that you asked a couple of colleagues to check the English of the manuscript. I believe the English is now fine. However, the readability of the manuscript - particularly in the Introduction - continues to be impaired by a lack of logical flow of information.

I have contacted PeerJ to see if they offered help with this kind of thing, but they do not. They suggested that I provide you with guidance. This is extremely difficult to do via a verbal description in a letter, so what I have done is downloaded the latest version of your manuscript, and using track changes, I have provided (many) suggestions about how to improve the wording of the manuscript, and also pointed out where there is missing information, and where the logical flow of that information goes astray. This feels like a very heavy-handed approach from an editor, however, I cannot think of another way forward given the writing is still not clear enough for publication.

I note that the senior author on this manuscript, Professor Fernandez, has around 200 publications and hence obviously has the skills to produce clearly-written manuscripts. I suggest that you also call on her help to help get the writing in the manuscript to a standard that is publishable - and also to help you address a few minor issues in the Results that I have noted that need addressing.

So, please refer to the specific comments in the pdf that I can send to you now via the PeerJ system. If you would like the docx version of this pdf (which may make life easier for you), please get in touch with me via email, and I can send it to you directly. I cannot work out how to send a docx to you via the PeerJ system.

With best wishes,

Genevieve

---

## Round 0.4 · Minor Revisions

Dear Sonia and team,

I must apologise for the delayed response to your re-submission. I have been unwell for well over a week, and it has been difficult to work.

You have made real headway with the clarity of the manuscript, and we are very close to acceptance. I have attached your latest doc, with (very) minor suggestions about how you might continue to clarify the manuscript. Please go through these suggestions to see what you would like to keep, and what you would like to adjust, and then send back to me. Unless you decide to make major changes (and I cannot see why you would), I am fairly confident that once you resubmit, I will be able to accept the manuscript, and you will move forward to the typesetting and proof-reading support.

BTW, I can only send you a pdf via the PeerJ system. If you want the docx version to save you time, email me, and I will send it to you directly.

Congratulations for all your hard work and resilience. I hope you and your team are doing OK in these pretty difficult times.

Genevieve

---

## Round 0.5 · accepted · Accept

Dear Sonia and co-authors,

Thank you for considering each of the suggested (minor) changes, and for explaining your decision to deal with some of those suggestions in a different manner to that suggested.

Congratulations on all your hard work, and I look forward to seeing your paper in print in PeerJ in the near future!

Best wishes,

Genevieve